# Slater–Pauling Behavior in Half-Metallic Heusler Compounds

**DOI:** 10.3390/nano13132010

**Published:** 2023-07-05

**Authors:** Iosif Galanakis

**Affiliations:** Department of Materials Science, School of Natural Sciences, University of Patras, 26504 Patra, Greece; galanakis@upatras.gr

**Keywords:** Heusler compounds, ab initio calculations, half-metals, magnetic materials, Slater–Pauling rule

## Abstract

Heusler materials have become very popular over the last two decades due to the half-metallic properties of a large number of Heusler compounds. The latter are magnets that present a metallic behavior for the spin-up and a semiconducting behavior for the spin-down electronic band structure leading to a variety of spintronic applications, and Slater–Pauling rules have played a major role in the development of this research field. These rules have been derived using ab initio electronic structure calculations and directly connecting the electronic properties (existence of spin-down energy gap) to the magnetic properties (total spin magnetic moment). Their exact formulation depends on the half-metallic family under study and can be derived if the hybridization of the orbitals at various sites is taken into account. In this review, the origin and formulation of the Slater–Pauling rules for various families of Heusler compounds, derived during these two last decades, is presented.

## 1. Introduction

Half-metallic magnets have garnered significant attention in the past two decades due to their potential applications in spintronics and magnetoelectronics [1,2]. Integrating the spin degree of freedom into conventional electronic devices based on semiconductors offers several advantages, such as non-volatility, increased data processing speed, reduced electric power consumption, and improved integration densities [3,4,5]. In Figure 1, a schematic representation of the density of states of a half-metallic magnet is presented. The spin-up band structures displays typical metallic characteristics, while the spin-down band structure behaves as a semiconductor with an energy gap. This gap results in 100% spin polarization at the Fermi level, enabling the possibility of achieving fully spin-polarized currents in these materials. This property maximizes the efficiency of magnetoelectronic devices [6,7,8].

The Heusler compounds, named after Heusler who first studied them [9], form a diverse family of materials exhibiting a vast variety of electronic behaviors [10,11]. Among them, several are magnetic being localized magnets, antiferromagnets, helimagnets, Pauli paramagnets, etc. [10,11,12,13,14,15,16,17,18]. The initial focus was on Heusler compounds that crystallize in the $L2_{1}$ structure, consisting of four face-centered cubic (fcc) sublattices (see Figure 2) and having the chemical formula $X_{2}YZ$. However, it was later discovered that one of the sublattices can be left unoccupied, resulting in what are commonly referred to as half- or semi-Heusler compounds having the chemical formula XYZ, whereas the $L2_{1}$ compounds are known as full-Heusler compounds. Some full-Heusler compounds (e.g., when *X* is Mn) prefer to crystallize in a tetragonal variation of the cubic $L2_{1}$ lattice [19,20]. One well-known semi-Heusler compound is NiMnSb [21]. In 1983, de Groot and collaborators demonstrated through first-principles electronic structure calculations that NiMnSb is actually a half-metallic magnet [22]. Following the discovery of de Groot et al., several semi- and full-Heusler compounds have been predicted to be also half-metallic magnetic [23,24,25,26,27]. Heusler half-metallic compounds are appealing for practical applications such as spin-injection devices [28], spin-filters [29], tunnel junctions [30], or giant magnetoresistance (GMR) devices [31,32] due to their relatively high Curie temperature, which exceeds by far room temperature for most of them [10,11], and due to their structural similarity to the zinc-blende and rocksalt structures, adopted by binary semiconductors widely used in industry (in Figure 2 when the C and D sites are empty the structure is the zinc-blende one and when the B and D sites are empty the structure is the rocksalt one). Other known half-metallic materials include certain oxides (e.g., CrO${}_{2}$ and Fe${}_{3}$O${}_{4}$) [33], manganites (e.g., La${}_{0.7}$Sr${}_{0.3}$MnO${}_{3}$) [33], double perovskites (e.g., Sr${}_{2}$FeReO${}_{6}$) [34], pyrites (e.g., CoS${}_{2}$) [35], transition metal chalcogenides and pnictides (e.g., CrSe and CrAs) [36,37,38,39,40,41,42,43,44], europium pnictides (e.g., EuN) [45], diluted magnetic semiconductors (e.g., Mn impurities in Si or GaAs) [46,47], and d${}^{0}$ ferromagnets like CaAs [48]. Thin films of CrO${}_{2}$ and La${}_{0.7}$Sr${}_{0.3}$MnO${}_{3}$ have been found to exhibit almost 100% spin polarization at the Fermi level at low temperatures [33,49].

Slater and Pauling had shown in two pioneering papers that in the case of binary magnetic compounds, the spin-up *d*-valence states are completely occupied and when one adds one extra valence electron, this occupies spin-down states only and the total spin magnetic moment decreases by about 1 $\mu_{B}$ [50,51]. Interestingly, a similar behavior can also be found in half-metallic magnetic Heusler compounds as confirmed by first-principles (ab-initio) electronic structure calculations. For these materials, the spin-down band-structure is fixed; the number of spin-down occupied bands and their character does not change among the half-metallic members of the same family of Heusler compounds. The extra valence electron now occupies exclusively spin-up states, increasing the total spin magnetic moment by about 1 $\mu_{B}$. Here, I should note that the spin-down character is by default assigned to the electrons for which the band-structure presents the energy band gap at the Fermi level, irrespective of whether they are the majority- or minority-spin electrons. Initially, the semi-Heusler compounds like NiMnSb were studied and the total spin magnetic in the unit cell, $M_{t}$ scales, as a function of the total number of valence electrons in the unit cell, $Z_{t}$, following the relation $M_{t}=Z_{t}-18$ [23]. A similar relation $M_{t}=Z_{t}-24$ was derived then also for the half metallic full-Heusler compounds [24]. Later, similar types of expressions have been derived for a series of other families of Heusler compounds: the inverse Heuslers [52] and the ordered quaternary Heuslers [53,54]. Also, generalized versions of these rules stand for the disordered quaternary and quinternary Heusler compounds [25,55,56] and when one transits from the semi- to the full-Heuslers by populating the empty site [57].

The Slater–Pauling (SP) rules in half-metallic Heusler compounds, as established and confirmed by ab initio calculations, connect the electronic properties (appearance of the half-metallic behavior) directly to the magnetic properties (total spin magnetic moments) and thus offer a powerful tool to the study of half-metallic compounds since (i) magnetic measurements can be used to confirm the half-metallic character of a compound, and (ii) simple valence electrons counting can predefine the magnetic properties of a half-metal. The aim of the present review is to provide an overview of the origin of the SP rule in all aforementioned families of half-metallic Heusler magnets as derived from first-principles electronic band structure calculations based on the density functional theory.

## 2. Semi-Heusler Compounds

I will start the discussion from the semi-Heusler compounds like NiMnSb, since they were the first to be predicted to be half-metals [22]. They have the general formula XYZ, where *X* and *Y* are transition metal atoms and *Z* is a metalloid. As mentioned above, the number of the spin-down valence bands in the half-metallic semi-Heusler compounds is a constant integer, since there is a gap and all spin-down bands below the Fermi level are fully occupied, and equal nine. In the lower panel of Figure 3 I schematically present the character of the bands and their degeneracy in the spin-down band structure. The bands below the Fermi level are occupied while the ones above it are empty. The *sp* element provides in the spin-down electronic band structure a single *s* and a triple-degenerate *p* band deep in energy; they are located below the *d*-states and accommodate *d*-charge from the transition metal atoms. The *d*-orbitals of the two transition metal atoms hybridize strongly creating five occupied bonding and five unoccupied antibonding *d*-states in the spin-down band structure as shown for NiMnSb compounds in the upper panel of Figure 3 [23]; note that $d_{1},d_{2},d_{3}$ correspond to the $d_{xy},d_{yz},d_{xz}$ and $d_{4},d_{5}$ correspond to the $d_{x^{2}+y^{2}},d_{z^{2}}$ orbitals. Each set of five occupied(unoccupied) *d*-hybrids contains the double degenerate $e_{g}$ and the triple degenerate $t_{2g}$ states. Note that the notations $e_{g}$ and $t_{2g}$ are, strictly speaking, valid only for states at the center of the Brillouin zone; however, the energy bands formed by them are energetically rather separated, and therefore this notation can be used to describe the different bands [23]. As a result, there are in total exactly nine occupied spin-down states per unit cell. The total spin magnetic in the unit cell $M_{t}$ (in $\mu_{B}$) equals the number of spin-up valence electrons minus the number of spin-down valence electrons or equivalently the total number of valence electrons $Z_{t}$ in the unit cell minus two times the number of spin down electrons. The latter expression, $M_{t}=Z_{t}-18$ for the half-metallic semi-Heusler compounds is the most popular form of the Slater–Pauling rule for these compounds.

The SP rule suggests that $M_{t}$ depends only on the total number of valence electrons in the unit cell and is independent of the chemical species of the constituent atoms, e.g., FeMnSb and CoCrSb have both 20 valence electrons and a total spin magnetic moment of 2 $\mu_{B}$. When the total spin magnetic moment is positive and thus there are more than 18 valence electrons in the unit cell, in the spin-up band structure all nine bonding *s*-, *p*- and *d*-states are occupied, as in the spin-down band, and the extra charge occupies the antibonding spin-up states [23]. In Figure 4, I present the ab initio calculated total spin magnetic moments versus the total number of valence electrons for some selected representative semi-Heusler compounds studied in Reference [23]. For most of the studied compounds, the total spin magnetic moments fall exactly on top of the line representing the SP rule. CoTiSb, which has exactly 18 valence electrons, is a well-known semiconductor. The compounds represented by the red spheres and with the yellow background are half-metals. Compounds like RhMhSb and NiMnSe slightly deviate from the perfect SP behavior since there is a spin-down energy gap but the Fermi level crosses either the valence or conduction spin down band. Finally, CoFeSb which has 22 valence electrons is the half-metallic compound with the maximum possible value of total spin magnetic moment. When an extra valence electron is added and $Z_{t}$ reaches 23, the half-metallic state is no more energetically favorable since otherwise five electrons would have to be accommodated in the antibonding spin-up bands.

A special case of semi-(half-) Heusler compounds are the ones containing alkali metals. Recently, Thuy Hoang and collaborators studied in detail such compounds possessing the chemical formula *A*Cr*Z*, where *A* is an alkali metal (Li, Na or K) and *Z* is P, As or Sb [58]. It was shown that they are also half-metals following a $M_{t}=Z_{t}-8$ variant of the SP rule. In their case, the spin-down band structure contains exactly four fully-occupied bands. Thuy Hoang et al. have shown that in the case of the alkali-based semi-Heusler compounds, the occupancy of the fours sites shown in Figure 2 is as follows: *Z* atoms at site A (black spheres), *A* atoms at site B (pink spheres), and Cr atoms at the D site (green spheres). The C-site (black squares) is empty. As a result, one should now consider first the interaction of the orbitals of the Cr and *Z* atoms, which are the nearest neighbors. Below the Fermi level in the spin-down band structure there is a single *s* band rising from the *s*-states of the *Z* atoms. This is followed by a band which is triple-degenerate at the Γ point. These bands are due to the hybridization between the *p* states of the *Z* atom and the $t_{2g}$ *d*-states of Cr, which transform following the same irreducible representation; the three bonding hybrids are below the energy gap and the three antibonding hybrids are above the gap. Thus, in total in the spin down band structure, there are exactly four occupied states.

## 3. Full-Heusler Compounds

In the case of the half-metallic $L2_{1}$ full-Heuslers (chemical formula $X_{2}YZ$) the origin of the SP rule is more complicated due to the more complicated hybridization effects between the *d*-orbitals [24]. Now there are exactly 12 spin-down occupied bands. In Figure 3, I present the hybridization scheme for the spin-down electronic band structure for the full-Heusler compounds. The *sp* element provides the spin-down electronic band structure with a single *s* and a triple-degenerate *p* band deep in energy as for the semi-Heusler compounds. With respect to the *d*-orbitals, one has to first consider the interaction between the *X* elements. Although the symmetry of the $L2_{1}$ lattice is the tetrahedral one, the *X* elements themselves, if one neglects the *Y* and *Z* atoms, form a simple cubic lattice and sit at sites of octahedral symmetry (see lattice in Figure 2) [24]. The *d*-orbitals of the neighboring *X* atoms hybridize, creating five bonding *d*-states and five antibonding *d*-states as shown schematically in the upper panel of Figure 3 where the case of Co${}_{2}$MnSi is presented. The first set of bonding orbitals afterwards hybridize with the *d*-orbitals of the *Y* atoms (Mn in our case), creating five occupied and five unoccupied *d*-hybrids (two of $e_{g}$ and three of $t_{2g}$ character). The second set of the X−X orbitals cannot hybridize with the *Y d*-orbitals since the former do not obey the tetrahedral symmetry but the more general octahedral symmetry. These five hybrids are located exclusively at the *X* atoms and are usually called in the literature non-bonding hybrids [24]. They split in the triple-degenerate $t_{1u}$ and double-degenerate $e_{u}$ states. As shown in the lower panel of Figure 3 only the spin-down $t_{1u}$ are occupied in the half-metallic full Heusler compounds leading to a total of 12 occupied spin-down states and the SP relation is now $M_{t}=Z_{t}-24$ [24].

In the case of full-Heusler compounds, when $Z_{t}>24$ is valid, the spin-up non-bonding $e_{u}$ states are the first to be occupied followed by the antibonding states, while when $Z_{t}<24$ the Fermi level crosses either the spin-up non-bonding $t_{1u}$ states or the spin-up bonding *d*-states [24]. In Figure 5, I have gathered some representative cases of ab initio calculated half-metallic full-Heusler compounds calculated in Reference [24]. Fe${}_{2}$VAl, which has 24 valence electrons, is a well-known semiconductor. The total spin magnetic moment reaches a maximum value of 5 $\mu_{B}$ for Co${}_{2}$MnSi, which has 29 valence electrons per unit cell. Although standard electronic band structure calculations show that for Co${}_{2}$FeSi the Fermi level falls above the spin-down energy gap, calculations treating it as a strongly-correlated system restore half-metallicity and its total spin magnetic moment reaches a value of 6 $\mu_{B}$, [59]. Finally, the compounds with less than 24 valence electrons like Mn${}_{2}$VGe and Mn${}_{2}$VAl are half-metallic ferrimagnets and the V spin magnetic moment is antiparallel to the spin magnetic moment of the Mn atoms [60].

## 4. Inverse Heusler Compounds

The $L2_{1}$ lattice of the full-Heusler compounds occurs when the valence of *X* is larger than the valence of the *Y* transition metal atom. Most of the compounds where the opposite occurs and the valence of *X* is smaller than the valence of the *Y* atom crystallize in the so-called XA lattice and are known as inverse Heusler compounds. The sequence of the atoms is now X-X-Y-Z as shown in Figure 2 and the prototype compound is Hg${}_{2}$TiCu. Several inverse Heuslers have been studied using first-principles electronic structure calculations in the literature [61,62,63,64,65]. Experimentally, the XA lattice structure of the inverse Heusler compounds has been confirmed by experiments on Mn${}_{2}$CoGa and Mn${}_{2}$CoSn films as well as Co doped Mn${}_{3}$Ga samples [66,67,68,69]. Also, Cr${}_{2}$CoGa has been found to crystallize in the XA structure and posses a very high Curie temperatures exceeding the 1000 K [70].

Using ab initio calculations, a large number of possible inverse Heuslers have been studied in Reference [52]. The ab initio calculated total spin-magnetic magnetic moments versus the total number of valence electrons are shown in Figure 6. Several of the compounds in Reference [52] have been identified as half-metallic magnets, but now there are three distinct variants of the SP rule depending on the hybridization scheme shown in Figure 3. When *X* is Sc or Ti, the total spin magnetic moment per formula unit (or unit cell) in $\mu_{B}$ follows the SP rule $M_{t}=Z_{t}-18$. When *X* is Cr or Mn, the SP variant is $M_{t}=Z_{t}-24$. For the *X* = V case, the form of the SP was found to be material specific. The occurrence of these rules can be explained using simple hybridization arguments of the transition metal *d*-orbitals as shown in Figure 3. In fact, when *X* is Cr or Mn the situation is similar to the usual Heusler compounds discussed above, but when X is Sc or Ti, the Fermi level in the minority-spin band structure is located below the non-bonding $t_{1u}$ states and now only nine spin-down states are occupied and the SP rule becomes $M_{t}=Z_{t}-18$. Thus, the origin of the latter has no relation to the case of the semi-Heuslers, which obey the same SP rule. The third variant of the Slater–Pauling rule occurs when X is Cr or Mn and Y is Cu or Zn. In this case, Cu or Zn *d*-states are completely occupied being in energy far below the *X d*-states as shown in the lower panel of Figure 3. The *d*-states of the neighboring *X* atoms hybridize, creating five bonding and five antibonding *d* states. Thus, for half-metals, the occupied spin-down band contain the five *d*-states of the Cu or Zn, the *s* and *p* states of the sp atom and the five bonding *d*-states formed due to the interaction of the orbitals of the *X* atoms. This in total there are 14 occupied spin-down bands and these half-metallic compounds follow a $M_{t}=Z_{t}-28$ SP rule.

## 5. Ordered Quaternary Heusler Compounds

Except for the usual and inverse full-Heusler compounds, another full-Heusler family is the Ordered Quaternary Heuslers (LiMgPdSn-type ones); also known as LiMgPdSb-type Heusler compounds [71]. These are quaternary compounds with the chemical formula $(XX^{\prime })YZ$ where *X*, $X^{\prime }$ and *Y* are transition metal atoms. The valence of $X^{\prime }$ is lower than the valence of *X* atoms, and the valence of the *Y* element is lower than the valence of both *X* and $X^{\prime }$. The sequence of the atoms along the fcc cube’s diagonal is $X-Y-X^{\prime }-Z$ as shown in Figure 2, which is energetically the most stable [72]. The correct notation is $(XX^{\prime })YZ$ since the *X* and $X^{\prime }$ atoms occupy the A and C sites, which have the same local symmetry but usually the parenthesis is omitted. Although early studies concerned only a few LiMgPdSn-type half-metallic compounds [71,73], recent studies have been devoted to the study of a large series of such compounds [53,54,74,75].

More precisely, in Reference [53], the first-principle’s electronic structure calculations have been employed to study the electronic and magnetic properties of 60 LiMgPdSn-type ordered quaternary Heusler compounds, and in Reference [54] a large number of quaternary Heusler compounds including 4*d* and 5*d* chemical elements have been studied. It was shown that most of these compounds were also half-metals obeying the same Slater–Pauling rule for full-Heusler compounds, $M_{t}$ = $Z_{t}$ − 24, with only few exceptions, which follow the $M_{t}$ =$Z_{t}$ − 18 variant. The driving force behind these two SP rules is similar to the one in inverse Heuslers as shown in Figure 3. In Figure 7, I have gathered the ab initio calculated total spin magnetic moment in the unit cell versus the total number of valence electrons for several ordered quaternary Heusler compounds studied in References [53,54]. There are several compounds following the $M_{t}$ = $Z_{t}$ − 24 rule, which have less than 24 valence electrons and are half-metallic ferrimagnets. The most interesting case is CrVTiAl, which falls on top of the $M_{t}$ = $Z_{t}$ − 18 variant and which has a total of 18 valence electrons and thus should have a zero total spin magnetic. Interestingly, this compound has been predicted to be a fully-compensated ferrimagnetic half-metallic compound and not a simple semiconductor [76]. This prediction has been confirmed experimentally [77]. Finally, it should be noted that the maximum total spin magnetic moment is achieved by CoIrMnSb, which has 30 valence electrons per unit cell like Co${}_{2}$FeSi.

## 6. Disordered Heusler Compounds

Starting from a full-Heusler compound, one can derive three distinct families of disorderd quaternary Heusler compounds having the chemical formulas ${\left(X_{x}X_{1-x}^{\prime }\right)}_{2}YZ$, $X_{2}\left(Y_{y}Y_{1-y}^{\prime }\right)Z$ and $X_{2}Y\left(Z_{z}Z_{1-z}^{\prime }\right)$ as shown in Figure 2. These disordered compounds behave exactly like the usual full-Heusler compounds. They are half-metallic magnets when the two extreme compounds are also half-metals and obey the $M_{t}$ = $Z_{t}$ − 24 SP rule [25,55]. Ozdogan et al. have shown that also the quinteranry Heusler compounds with the chemical type $X_{2}\left(Y_{y}Y_{1-y}^{\prime }\right)\left(Z_{z}Z_{1-z}^{\prime }\right)$ are half-metals following the same SP rule as the usual full-Heuslers [56].

An interesting case is when one starts from a semi-Heusler compound and starts populating the vacant site to end up with a full-, inverse- or ordered-quaternary-Heusler compound obeying the $M_{t}$ = $Z_{t}$ − 24 SP rule, e.g., Co${}_{1+x}$MnSi, Mn${}_{2}$Co${}_{x}$Si and CoFe${}_{x}$MnSi, where 0≤x≤1. As it was shown in Reference [57], in the case of such half-metallic magnetic compounds, a generalized version of the SP rule is valid $M_{t}=Z_{t}-\left(18+6x\right)$, so that for the end perfect compounds (x=0 or x=1), one regains the well-known $M_{t}$ = $Z_{t}$ −18 and $M_{t}$ = $Z_{t}$ − 24 SP rules.

## 7. Summary and Outlook

Half-metallic magnets have attracted significant interest in the field of spintronics and magnetoelectronics due to their predicted perfect electrons spin-polarization at the Fermi level. Among them, half-metallic Heusler compounds are of particular interest due to their structural similarity to the binary semiconductors and their very high Curie temperatures. Slater–Pauling rules are a strong tool in their study since they directly link their electronic properties (number of valence electrons in the unit cell) to their magnetic properties (total spin magnetic moment in the unit cell). The Fermi level is situated within the spin-down energy gap, and the total number of occupied spin-down states is fixed. The additional electrons exclusively occupy spin-up states. Thus, these rules are of particular importance in studying the half-metallic Heusler compounds for two reasons: (i) magnetic measurements can be utilized to confirm the half-metallic nature of samples, and (ii) the total spin magnetic moment of a half-metallic compound can be predetermined based on its chemical formula.

The origin of the Slater–Pauling rule varies among different compound families under investigation. In each case, the hybridization of orbitals must be considered, accounting for both local and crystal symmetry. In semi-Heusler compounds like NiMnSb, the hybridization of *d*-orbitals results in a relation of $M_{t}=Z_{t}-18$, where $M_{t}$ represents the total spin magnetic moment in $\mu_{B}$ and $Z_{t}$ denotes the total number of valence electrons in the unit cell. However, in the case of full-Heuslers such as Co${}_{2}$MnSi, the presence of *d*-hybrids with octahedral symmetry exclusively at the Co sites leads to a more complex hybridization effect and a SP rule of $M_{t}=Z_{t}-24$. Inverse and ordered quaternary Heusler compounds present a variety of SP rules depending on the exact position of the Fermi level with respect to the spin-down states. Finally, the SP rule survives in the case of the disordered quaternary Heusler compounds and the doped semi-Heuslers.

## Figures and Tables

**Figure 1 nanomaterials-13-02010-f001:**
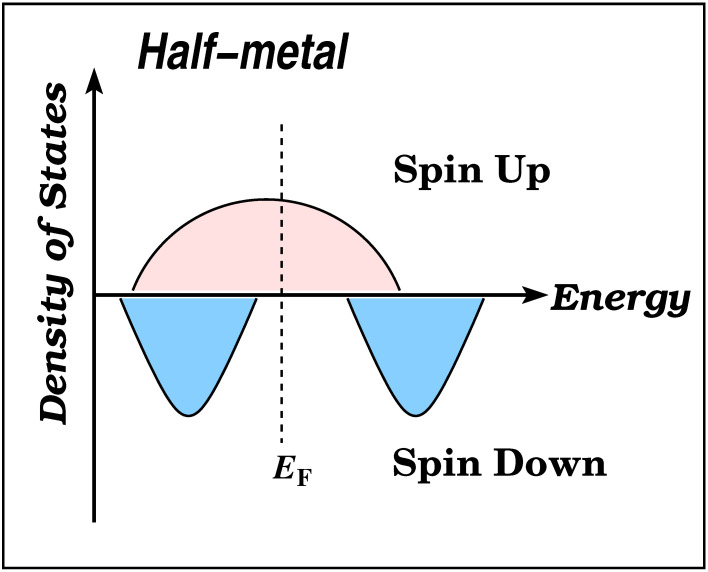
Schematic representation of the density of states (DOS) of a half-metallic material. The positive DOS values correspond to the spin-up electronic band structure, which shows a metallic behavior. The negative DOS values correspond to the spin-down electrons, which exhibit a semiconducting behavior.

**Figure 2 nanomaterials-13-02010-f002:**
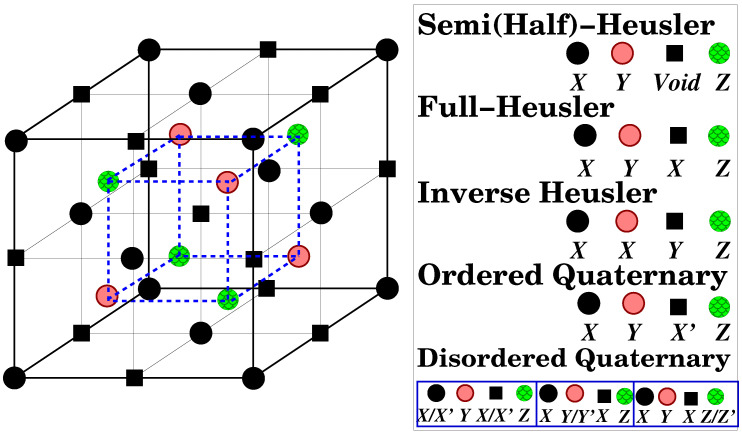
Schematic representation of (i) the $C1_{b}$ cubic structure adopted by the semi(half)-Heusler compounds, (ii) the $L2_{1}$ structure adopted by the full-Heusler compounds, (iii) the XA structure adopted by the inverse Heusler compounds, (iv) the Y structure adopted by the ordered quaternary Heusler compounds, and (v) the various types of disordered quaternary Heusler compounds. The black spheres, pink spheres, black squares and green spheres are widely called A, B, C and D sites, respectively. The large cube in the figure contains exactly four unit cells.

**Figure 3 nanomaterials-13-02010-f003:**
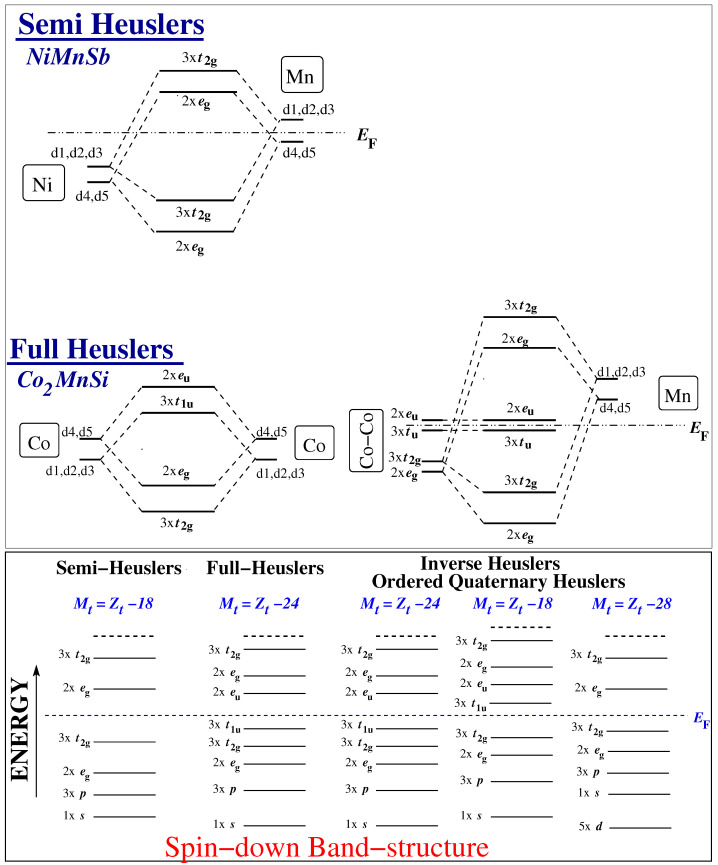
Upper Panel: Schematic representation of the hybridization schemes between the *d* orbitals of transition metal atoms in semi- and full-Heusler compounds. Lower panel: Schematic representation of the energy levels of the spin-down electronic band structure for all four ordered families of half-metallic Heusler compounds under study. Below the Fermi level are located the occupied states. Numbers in front of orbitals denote the corresponding degeneracy. For the definition of the orbitals, see text.

**Figure 4 nanomaterials-13-02010-f004:**
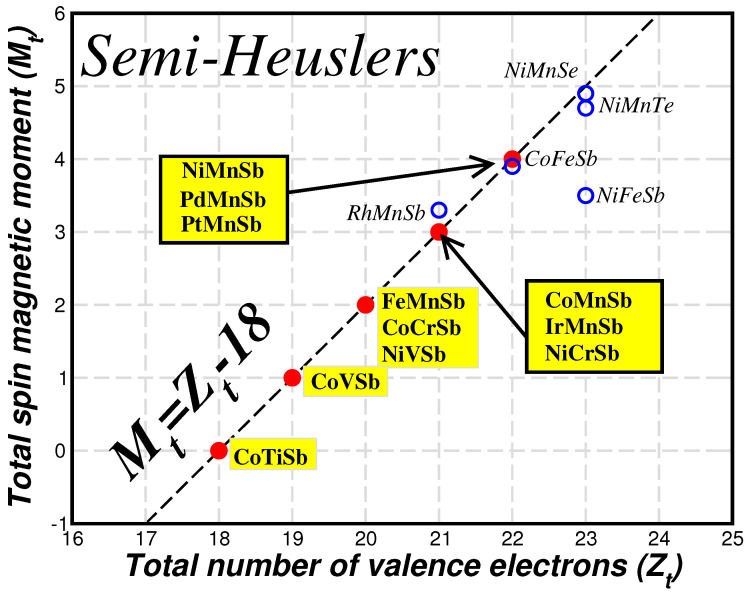
Ab-initio calculated total spin magnetic moments, $M_{t}$, in $\mu_{B}$ as a function of the total number of valence electrons, $Z_{t}$, in the unit cell for selected representative semi-Heusler compounds. The dashed line represents the $M_{t}=Z_{t}-18$ SP rule. Within the yellow background are the compounds that are perfect half-metals. Data are taken from Reference [23].

**Figure 5 nanomaterials-13-02010-f005:**
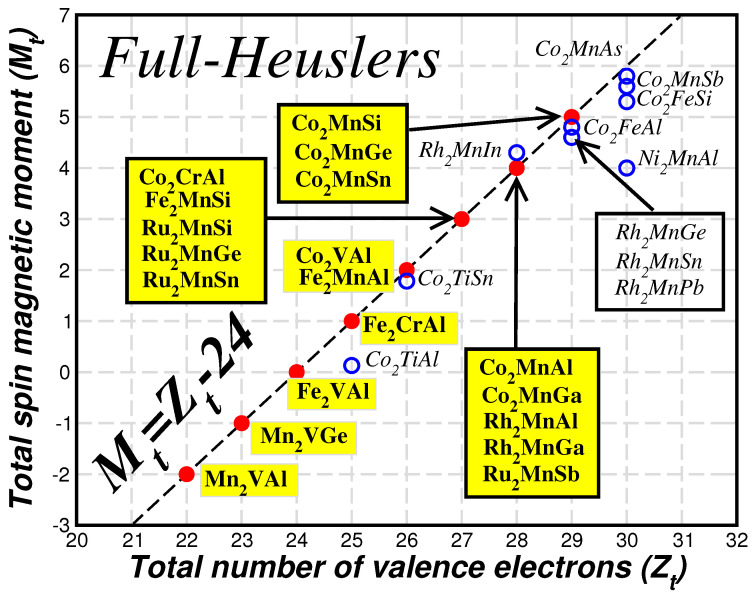
Similar to Figure 4 for the full-Heusler compounds. Data are taken from Reference [24].

**Figure 6 nanomaterials-13-02010-f006:**
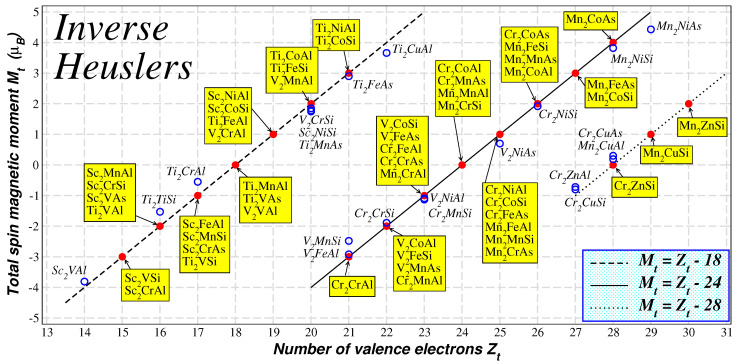
Similar to Figure 4 for the inverse Heusler compounds. The three lines represent the three variants of the Slater–Pauling rule for these compounds. Data are taken from Reference [52].

**Figure 7 nanomaterials-13-02010-f007:**
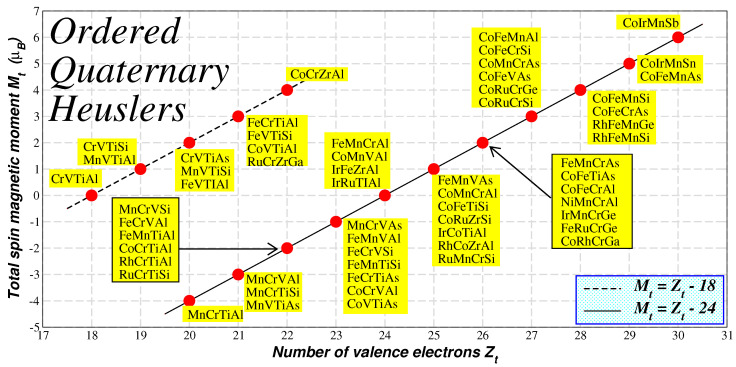
Similar to Figure 4 for the Ordered Quaternary Heusler compounds. The two lines represent the two variants of the Slater–Pauling rule for these compounds. Data are taken from References [53,54].

## Data Availability

The data presented in this study are available on request from the author.

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
