# Peer review of "Slater–Pauling Behavior in Half-Metallic Heusler Compounds"

_nanomaterials, 2023, doi:10.3390/nano13132010_

Round 1
Reviewer 1 Report
Report on:
Slater-Pauling behavior in half-metallic Heusler compounds
by: Iosif Galanakis
The author reviews the theory on halfmetallic Heusler compounds and the relation to the Slater Pauling rule. The review is well written and should be published in its present form with some small additions.
The Half-metal is well explained with Figure 1, however, the Slater-Pauling behaviour and its general background should be also introduced in a simple
manner.
In Figures 4 to 7, those compounds that have been confirmed by experiments to exist should be marked in a different colour. This might be helpful to conduct more experiments.
Some work of J. Kübler, who was the first to explain the magnetic behaviour of Heusler compounds, should be cited, and maybe there are also some other interesting publications on the Slater-Pauling behavior and Heusler compounds behavior available.
Author Response
I would like to thank the Referee for his/her appreciation regarding my review article.
The referee has risen three points.
1) In order to make the explanation of the origin of the Slater-Pauling rules more understandable, I have expanded figure 3 adding a panel which shows schematically the various hybridization schemes between the d-orbitals of neighboring atoms. I refer to it whenever I discuss the hybridization procedure.
2) Although the suggestion of the Referee seems reasonable, the amount of publications containing experiments is huge. Most of them concern thin films, multilayers and other low-dimensional structures. Thus one should not just include the names of the compounds which exist experimentally, but also discuss their structural properties. Such a presentation by far exceeds the scope of the present review.
3) Although, thousands of articles are devoted to Heusler compounds, the article of 1983 in Physical Review B by Kubler, William and Sommers is one of the fundamental articles explaining their magnetic properties. I want to thank the Referee for bringing thιs ommision to my attention. I have added this article in the reference list (reference 12) as well as two more recent publications of Kubler on the Mn-based full Heusler compounds (references 19 and 20)
Reviewer 2 Report
In this paper the origin and formulation of the Slater-Pauling rules for various families of Heusler alloys is presented. I think that the paper will very useful for readers, especially for scientists which work with Heusler alloys. But I have two remarks.
1. Authors say in abstract and in text that "... magnets which present a metallic behavior for the spin-up and a semiconducting for the spin-down electronic band structure...". I know that the situation can be inverse. I think that need to say "... magnets which present a metallic behavior for the one spin channel and a semiconducting for the another spin channel...".
2. The alkali based half-Heusler alloys exist also. They do not menshioned in the paper. The Slater-Pauling rule for them also need to present in the paper.
After these corrections the paper can be published as is.
Author Response
I would like to thank the Referee for his/her appreciation towards my manuscript. The referee has risen two points.
1) I believe there is a misunderstanding. It is correct that the energy-gap can be located either at the majority or minority spin band structure. Unlike the majority-spin(minority-spin) characterization, there is a freedom when one characterizes the bands of being of spin-up or spin-down character, In order to have the correct signs in the SP rules one should always demand the gap to be present in the spin-down band structure which is the minority spin band structure if the total spin magnetic moment is positive or the majority spin band structure if the total spin moment is negative. In some publications they assign the spin-up character to the majority-spin electrons and the spin-down to the minority spin electrons by default. I have added a new sentence (lines 61-64) to make clear how the spin-up(down) character of the bands is assigned.
2) Following the suggestion of the Referee, I have added at the end of section II on semi-Heusler compounds a paragraph discussing the alkali-metal-based Heusler compounds based mainly on a very recent article by Thuy Hoang et al. (reference 58)
Reviewer 3 Report
The review of Prof. Iosif Galanakis is brief and very clear overview of origin of the Slater-Pauling rule in half-metallic Heusler magnets. That is its undoubted merit. In fact, it is a variant of article by S. Nepal, R. Dhakal, I. Galanakis, S. M. Winter, R. P. Adhikari, and G. C. Kaphle / Phys. Rev. Materials, 2022, v.6, 114407 (https://doi.org/10.1103/PhysRevMaterials.6.114407), supplemented by K.Özdogan, E.Sasıoglu, I.Galanakis / J. Appl. Phys. 2013, v.113, 193903 (http://dx.doi.org/10.1063/1.4805063).
However, the articles in Phys. Rev. Materials and J. Appl. Phys. are only available by subscription, and Nanomaterials is a highly rated open access journal, which is valuable for a wide scientific audience. Therefore, such an overview will be welcomed with interest by readers of Nanomaterials. Especially since the review is based on simple and purely qualitative reasoning about the structure of molecular orbitals and a specific discussion of the effects of hybridization of d-orbitals.
My critical comments are the following. Keywords "Ab-initio calculations" are present in the beginning of the article. These words are also found somewhere in the text of the article. However, there are no ab-initio calculations in this article. There are no results of DFT calculations by DOS (spin-polarized total density of states). It is necessary to give the characteristic spin-polarized DOS for each type of compounds under discussion. There is no description of quantum-chemical packages used in the ab-initio calculations. The author used a full-electron DFT calculation method. It is unclear what are the parameters of the calculations: the choice of unit cell, the number of points in k-space, cutoff energies of wave functions and electron densities. This material is important for evaluation and reproducibility of the results. All of the above mentioned should definitely be added to the article as a separate section, so that the paper is self-sufficient and verifiable.
Author Response
I would like to thank the Referee for his/her report and for his/her appreciation towards mymanuscript.
The keyword "ab-initio" refers to the fact that all total spin magnetic moments presented in the manuscript have resulted from ab-initio electronic band structure calculations. I have not included any DOS pictures except the the schematic figure 1, because the scope of this review is to concentrate on the Slater-Pauling behavior of these compounds and not to present an extensive overview of the electronic properties of Heusler compounds. The reason is that the various SP rules have appeared in articles which I have coauthored spanning a period of 20 years (and not just the two articles mentioned by the Referee). They are very useful especially for experimentalists and no review exclusively focusing on them exists.
Also regarding the details of the calculations., since the presented results have been obtained by a variety of electronic structure codes it is not very helpful for the readers to give all the details. Under each figure there is reference to the article where these calculated values have appeared for the first time. And as stated at the end of the article in the Data Availability Statement all data are available on request from the author.
Round 2
Reviewer 3 Report
No comments.